# Investigating the Eye as a Biomarker of Gulf War Illness: Sphingolipid and Eicosanoid Composition in Tears and Plasma

**DOI:** 10.3390/biom15121716

**Published:** 2025-12-10

**Authors:** Laura Beatriz Paule Jimenez, Amanda Prislovsky, Loralei Ann Parchejo, Kimberly Cabrera, Andrew J. Nafziger, Daniel J. Stephenson, Charles E. Chalfant, Kristina Aenlle, Nancy Klimas, Fei Tang, Nawajes Mandal, Anat Galor

**Affiliations:** 1University of Miami Miller School of Medicine, Miami, FL 33125, USA; lbp49@miami.edu; 2Department of Ophthalmology, Hamilton Eye Institute, The University of Health Science Center, Memphis, TN 38163, USA; aprislo1@uthsc.edu; 3Research Services, Memphis VA Medical Center, Memphis, TN 38104, USA; 4Surgical and Research Services, Miami Veterans Affairs Medical Center, Miami, FL 33125, USA; loraleiann.parchejo@va.gov (L.A.P.); kimcabrera1@gmail.com (K.C.); kaenlle@nova.edu (K.A.); nklimas@nova.edu (N.K.); fei.tang@va.gov (F.T.); 5Departments of Medicine and Cell Biology, University of Virginia School of Medicine, Charlottesville, VA 22903, USA; ajn7pa@virginia.edu (A.J.N.); stephensondj@alumni.vcu.edu (D.J.S.); krj2sf@uvahealth.org (C.E.C.); 6Research Service, Richmond VA Medical Center, Richmond, VA 23298, USA; 7Institute for Neuro-Immune Medicine, Nova Southeastern University, Fort Lauderdale, FL 33328, USA; 8Department of Anatomy and Neurobiology, The University of Health Science Center, Memphis, TN 38163, USA; 9Bascom Palmer Eye Institute, University of Miami, Miami, FL 33125, USA

**Keywords:** gulf war illness, biomarkers, lipidomics, sphingolipids, eicosanoids, tears, plasma, inflammation, ocular surface

## Abstract

Gulf War Illness (GWI) is a chronic multi-symptom condition affecting veterans of the 1990–1991 Gulf War, with ocular discomfort increasingly recognized among its manifestations. This pilot study evaluated whether lipid alterations in tears and plasma could serve as potential biomarkers of GWI. Participants included Gulf War-era veterans seen in the Miami Veterans Affairs Hospital eye clinic from 2018–2022. Cases met GWI criteria, while controls were non-deployed, age- and gender-matched veterans without GWI. Participants completed systemic and ocular symptom questionnaires, and lipidomic profiling of tears and plasma quantified sphingolipids and eicosanoids. Compared to controls (n = 21), GWI cases (n = 19) reported greater ocular symptom burden, while ocular signs were similar between groups. Lipidomic analyses revealed increased tear eicosanoids ((±)14(15)-EET and (±)8(9)-EET), elevated plasma sphingomyelins (SM C16:0 DH, SM C20:0, SM C22:0), and reduced plasma monohexosylceramide (MHC C16:0) and sphingomyelin (SM C14:0) in cases. Logistic regression and random forest models identified plasma SM C16:0 DH and SM C20:0 as top predictors distinguishing GWI cases from controls, with an area under the receiver operating characteristic curve (AUC) of 0.89. These findings suggest lipid dysregulation in ocular and systemic compartments and support further investigation of tears as a minimally invasive source for biomarker discovery.

## 1. Introduction

Upon returning from the 1990–1991 Gulf War (GW), approximately one-third of GW Veterans developed a chronic, multi-symptom condition known as Gulf War Illness (GWI) [1,2]. GWI symptoms span complaints in multiple compartments including cognitive dysfunction, respiratory issues, musculoskeletal pain, irritable bowel syndrome, sleep disturbances, fatigue, and headaches [1,2]. In addition, prior studies, including our own, have found that ocular symptoms are common in GW Veterans. In a 1998 survey of 1844 Kansas Veterans, aged 24 to ≥50 at the time of the survey, the prevalence of “eyes very sensitive to light” (25% vs. 11%, OR: 2.62, 95% CI: 1.84–3.74) and “blurred or double vision” (13% vs. 5%, OR: 2.49, 95% CI: 1.55–4.00), with onset since 1990, was significantly higher among GW-deployed Veterans compared to non-deployed GW-era Veterans, even after adjusting for age, sex, income, and education level [3]. In our retrospective study, we found that individuals with GWI had more frequent dry eye disease (DED) symptoms compared to GW-era Veterans without GWI (50% vs. 33%; *p* = 0.04) [4]. These results were supported by a subsequent prospective study of 71 GW-era Veterans, in which those meeting GWI criteria had more severe dry eye symptoms (Ocular Surface Disease Index [OSDI], 41.20 ± 22.92 vs. 27.99 ± 24.03; *p* = 0.01) and higher ocular pain scores (2.63 ± 2.72 vs. 1.22 ± 1.50; *p* = 0.03, 0–10 Numeric Rating Scale [NRS]) compared to those without GWI [5]. These findings suggest that the eye may serve as an indicator of systemic disease processes in GWI.

However, reliable biomarkers for GWI are lacking, and the underlying etiology of disease remains incompletely understood. One proposed mechanism suggests that deployment-associated exposures (e.g., pesticides, pyridostigmine bromide [PB], and nerve agents), coupled with stress and genetic predisposition, created a neuroinflammatory state manifesting as GWI symptoms [6,7]. In a study of 304 GW-deployed Veterans, those stationed in combat zones had significantly higher odds of meeting GWI criteria if they reported taking PB pills (OR: 3.50, 95% CI: 1.65–7.41) or being within 1 mile of an exploded surface-to-surface (SCUD) missile (OR: 3.07, 95% CI: 1.53–6.19) [8]. Supporting the neuroinflammatory state of GWI, neuroimaging studies using Positron Emission Tomography (PET) with [^11^C]PBR28, a radioligand for the 18 kDa translocator protein (TSPO), have shown increased expression of TSPO, a marker of neuroinflammation, in Veterans with GWI [9]. Animal models have also reinforced the neuroinflammatory basis of GWI. In one study, adult male rats were exposed daily for four weeks to low doses of Gulf War Illness-related chemicals (GWIRCs), including PB, and the pesticides N,N-diethyl-meta-toluamide (DEET) and permethrin (PER), along with mild restraint stress. Six months post-exposure, analysis of the hippocampus noted decreased expression of the anti-inflammatory genes interleukin-4 (IL-4) and IL-10 and increased expression of the pro-inflammatory genes factor kappa B subunit 1 (NFκB1), IL-1α, IL-6, colony-stimulating factor 2 (Csf2), and B-cell lymphoma 6 (Bcl6) [10].

While brain tissue provides mechanistic insight, it is not a practical biomarker in humans, as it requires post-mortem samples or invasive procedures. Similarly, imaging techniques such as PET are expensive and impractical for widespread use as biomarkers in large populations. As a more accessible alternative, plasma inflammatory cytokines have been explored; however, they have not proven to be reliable markers for GWI due to inconsistencies across studies. For example, one study noted a trend towards elevated IL-4 and IL-13 among GW-era Veterans with neurological symptoms of blurred vision, tremors/shaking, balance problems/dizziness, and speech difficulty [11], whereas another study of 37 GWI subjects and 42 controls (Gulf War-era or deployment status not specified) found significantly lower mean levels of IL-4 and IL-13 in cases compared to controls (IL-4: 0.9 vs. 1.1 pg/mL; *p* = 0.04; IL-13: 8.2 vs. 9.0 pg/mL; *p* = 0.001) [12]. Similarly, discrepancies exist for other cytokines. One study comparing GWI subjects to GW-era controls reported significantly higher plasma IL-6 concentrations across multiple time points (*p* ≤ 0.03) [13], while another found significantly lower tumor necrosis factor alpha (TNF-α) levels in GWI compared to healthy controls (19.3 vs. 23.1 pg/mL; *p* < 0.0001) [12]. In contrast, a third study comparing 15 GWI Veterans to 33 healthy controls (including 8 Gulf War-era Veterans and 25 civilians) found no significant differences in IL-6 or TNF-α across groups (both *p* > 0.05) [9]. Such variability across studies highlights the difficulty of establishing reliable protein-based biomarkers for GWI.

Given these inconsistencies, there is a need for biomarkers that are specific, stable, accessible, and reflective of systemic disease activity. Lipid biomarkers have emerged as possible candidates in this regard. Metabolomic profiling has identified lipid pathways as central to the biochemical signature of GWI. In one analysis of 46 biochemical pathways, 8 were significantly altered, 78% of which involved lipids [14]. The same analysis noted elevated sphingolipids (SPL), including ceramides (Cer) and sphingomyelins (SM), in the serum of 20 GWI participants compared to 20 non-Veteran healthy controls. Additional work among GW-deployed Veterans (22 with GWI and 11 without) revealed broad disruptions in phospholipid (PL) profiles, with cases exhibiting elevated lysophosphatidylcholine (LPC), decreased saturated fatty acid-containing phosphatidylcholine (PC), and increased polyunsaturated fatty acid (PUFA)-enriched lysophosphatidylethanolamine (LPE) and ether LPE (eLPE) species [15]. Elevation of ω-6 arachidonic acid (AA) and ω-3 docosahexaenoic acid (DHA) within LPC and LPE was also observed in GWI cases compared to GW-deployed controls [15], which is noteworthy because AA is a precursor for pro-inflammatory eicosanoids, and DHA gives rise to anti-inflammatory mediators such as neuroprotectins and resolvins [16]. Uniquely, this study combined human and animal data, including two rodent models simulating GWI: a mouse model exposed to PB and permethrin (PER) and a rat model exposed to PB, PER, DEET, and chronic stress, demonstrating elevated LPC species in both GWI subjects and rodent models [16]. These shared lipid alterations highlight the translational relevance of findings across humans and animal models.

Beyond plasma, tear fluid may be a more practical source for examining disease biomarkers, given its greater accessibility. We have previously demonstrated the diagnostic value of SPL and eicosanoids in ocular surface disorders like Meibomian Gland Dysfunction (MGD) [17] and have also demonstrated their relevance in systemic disorders like post-traumatic stress disorder (PTSD) and depression [18], which are common comorbidities in GWI [19]. Building on this foundation, we conducted a pilot study comparing plasma and tear SPL and eicosanoid profiles in 40 GW-era Veterans to explore their potential relevance to disease and determine whether larger-scale studies are warranted. This work aims to advance our understanding of lipid abnormalities in GWI and contribute to the development of more precise and clinically meaningful biomarkers for disease characterization and management.

## 2. Materials and Methods

### 2.1. Study Design and Population

This pilot study included 40 Veterans evaluated at the Miami Veterans Affairs (VA) Hospital eye clinic between November 2018 and February 2022. Participants were included if they were active Veterans during 1990–1991, regardless of deployment status. Exclusion criteria encompassed individuals with unmanaged psychotic disorders lasting longer than six months without stable treatment, active infections or pregnancy, history of head trauma involving loss of consciousness exceeding five minutes, major neurological conditions unrelated to Gulf War Illness, and medical conditions interfering with study procedures. All participants provided written informed consent. The Miami VA Institutional Review Board approved the protocol, and the study was performed in accordance with the Declaration of Helsinki and the Health Insurance Portability and Accountability Act (HIPAA) regulations.

Cases were Veterans deployed to the Gulf who met criteria for Gulf War Illness, defined by both the Centers for Disease Control and Prevention (CDC) and modified Kansas definition. The modified Kansas definition required symptom onset within two years of deployment, fatigue after exercise as a predominant component, and allowance for stable comorbidities such as hypertension, diabetes, PTSD, depression, or mild traumatic brain injury. Controls were Gulf War-era Veterans who were not deployed to the Gulf, matched on age and gender, and who did not meet criteria for GWI under either definition.

### 2.2. Data Collection

Demographic information, comorbidities, medications, and medical and ocular diagnoses were collected for all participants. Given the multi-symptom nature of GWI, all individuals filled out standardized questionnaires regarding depression (Patient Health Questionnaire [PHQ] 9) [20], post-traumatic stress disorder (Post Traumatic Stress Disorder Checklist—Military version [PCL-M]) [21], insomnia (Pittsburgh Sleep Quality Index [PSQI]) [22], fatigue (Modified Fatigue Impact Scale [MFIS]) [23], and musculoskeletal symptoms (Widespread Pain Index [WPI]/Symptom Severity Scale [SS]) [24]. Quality of life was assessed with the 12-Item Short-Form Survey (SF-12) [25].

### 2.3. Clinical Examination

A number of validated questionnaires were used to capture ocular symptoms, including the Ocular Surface Disease Index (OSDI, range 0–100) [26], 5-Item Dry Eye Questionnaire (DEQ-5, range 0–22) [27], Neuropathic Pain Symptom Inventory modified for the Eye (NPSI-Eye, total score: range 0–100; sub-score range 0–10) [28], Convergence Insufficiency Symptoms Survey (CISS, 0–60) [29], and ocular pain intensity graded to a 0–10 Numeric Rating Scale (NRS). NRS scores were acquired for pain felt “right now”, “averaged over the last week”, and “worst over the last week”, as well as before and after placement of topical proparacaine hydrochloride 0.5%, the latter to assess potential central contributions to ocular pain [30].

Ocular surface examination findings captured, in the order assessed, ocular surface inflammation (InflammaDry [Quidel, San Diego], higher values indicate greater inflammation), tear stability (tear break-up time [TBUT], lower values indicate less stability), epithelial disruption (fluorescein corneal staining, higher values indicate more disruption), tear production (anesthetized Schirmer test at 5 min, lower values indicate less tear production), and eyelid and meibomian gland parameters (higher values indicate more abnormalities). InflammaDry was qualitatively graded based on the intensity of the pink stripe as none, mild, moderate, or severe. TBUT in seconds was measured three times in each eye after instilling 5 μL of fluorescein dye and values averaged. Corneal staining was captured by dividing the cornea into five areas and grading in each area on a scale of 0 = none to 3 = severe, with the scores summed. Eyelid vascularity was graded on a scale of 0 to 3 (0 none; 1 mild engorgement; 2 moderate engorgement; 3 severe engorgement) and meibum quality on a scale of 0 to 4 (0 = clear; 1 = cloudy; 2 = granular; 3 = toothpaste; 4 = no meibum extracted). Inferior meibomian gland dropout was graded to the Meiboscale based on Lipiscan (Johnson & Johnson, New Brunswick, NJ, USA) images [31]. The ocular surface examination was performed by a provider that was masked to the clinical symptoms for each patient.

### 2.4. Sample Collection

#### 2.4.1. Tear Collection

Tear samples were collected from the right and left eye using commercially available Schirmer strips (e.g., Bausch & Lomb, Rochester, NY, USA). Strips were left in place for five minutes after application of proparacaine, and samples were stored at −80 °C until protein extraction. For consistency, only the right eye sample was analyzed per patient. Blank Schirmer strips were evaluated during method development, and no analytes of interest were detected; therefore, routine analyses included only solvent blanks with and without internal standards at the beginning and end of each run to monitor background and carryover.

#### 2.4.2. Plasma Collection

Standard phlebotomy was used to obtain peripheral venous blood from each participant. Approximately 50 mL of blood was drawn into the following tubes: two 10 mL ethylenediaminetetraacetic acid (EDTA) lavender-top tubes (for plasma), two 10 mL green-top tubes (for plasma/heparinized plasma), and one 8.5 mL red-tiger-top serum separator tube (for serum). Blood collection was performed at the Miami VA, and all samples were processed and stored according to standardized protocols by Drs. Anelle and Klimas. About 200 μL of the plasma collected from the EDTA samples was used for lipidomic analysis (sphingolipids and eicosanoids).

### 2.5. Tear Biomarker Analysis

#### 2.5.1. Tear Sphingolipid Extraction and Analysis

Tear lipids were extracted from the full Schirmer strip using a modified Bligh and Dyer method, as previously described [17]. Total lipid recovery depended on the initial tear volume absorbed with the strip. Frozen samples were added to borosilicate glass tubes and spiked with 250 pmol of internal standards: ceramide-1-phosphate (C1P), sphingomyelin (SM), ceramide (Cer), monohexosylceramide (d18:1/12:0 species), sphingosine (So), sphinganine (Sa), sphingosine-1-phosphate (S1P), and sphinganine-1-phosphate (Sa1P) (d17:0 sphinganine/d17:1 sphingosine) (Avanti Polar Lipids). Methanol–chloroform (MeOH:CHCl3) (2:1) was then added, and the mixture was sonicated for ~2 min and incubated at 48 °C for 6 h. Extracts were centrifuged at 5000 rpm for 20 min, transferred to clean glass tubes, dried, reconstituted in 500 μL of methanol by sonication, centrifuged again at the same speed, and placed in injection vials for mass spectrometry analysis.

Sphingolipid analysis was performed using Ultra-Performance Liquid Chromatography (UPLC) coupled with Electrospray Ionization Tandem Mass Spectrometry (ESI-MS/MS) on a Shimadzu Nexera X2 LC-30AD system (Shimadzu Corporation, Kyoto, Japan) with a SIL-30AC auto injector and DGU-20A5R degasser. Separation was completed with an Acentis Express C18 column (5 cm × 2.1 mm, 2.7 μm) at 60 °C and a 0.5 mL/min flow rate, using an 8min reversed-phase LC method. This temperature was selected based on method development testing across 35–70 °C, which identified 60 °C as providing optimal peak shape and chromatographic separation for sphingolipids. The column was equilibrated with 100% Solvent A [methanol:water:formic acid (58:44:1, *v*/*v*/*v*) containing 5 mM ammonium formate] for 5 min followed by injection of a 10 μL sample. Elution began with 100% Solvent A (0–0.5 min), followed by a linear gradient to 100% Solvent B [methanol:formic acid (99:1, *v*/*v*) with 5 mM ammonium formate] between 0.5 and 3.5 min. Solvent B was kept at 100% from 3.5 to 6 min, reduced back to 0% from 6 to 6.1 min, and Solvent A returned to 100% from 6.1 to 8 min.

Mass spectrometry detection was performed on an AB Sciex Triple Quad 5500 (SCIEX, Framingham, MA, USA) using a targeted assay in multiple reaction monitoring (MRM) positive-ion mode. Quadrupole 1 (Q1) and Quadrupole 3 (Q3) of the mass spectrometer were set to monitor distinctive precursor and product ion pairs specific for sphingolipids [32], with fragmentation in Quadrupole 2 (Q2) induced by nitrogen collision gas. Instrument parameters were as follows: curtain gas: 30 psi; Collisionally Activated Dissociation (CAD): medium; ion spray voltage: 5500 V; temperature: 500 °C; gas 1: 60 psi; gas 2: 40 psi. Declustering potential, collision energy, and cell exit potential were varied for each transition. Full MRM transitions and instrument parameters for sphingolipids are provided in the Appendix A.

SPL species were identified by retention time and *m*/*z* ratio and quantified semi-quantitatively (pmol) from peak areas relative to internal standards, as described previously [32,33]. Quantitation was performed using a semi-quantitative approach commonly applied in targeted lipidomics workflows. A representative internal standard was added for each sphingolipid class, and a labeled internal standard was added for each eicosanoid at known concentrations. Analyte concentrations were derived from the ratio (area of analyte)/(area of internal standard) = (concentration of analyte)/(concentration of internal standard), since all values except the analyte concentration were known. This strategy allows for reliable quantitation across a large panel of lipid species for which generating individual calibration curves is not feasible and is standard practice in semi-quantitative LC–MS/MS lipidomics assays. Data were expressed as total pmol and as mole% to normalize for tear volume. Positive detection cutoffs were >100 signal to noise ratio (S/N) for Cer, SM, and hexosylceramide (Hex-Cer) and >10 S/N for low-expressing Sa, So, and S1P. Non-detectable values were reported as zero or “Non-Detectable (ND).” The method selectively measured non-dihydro species of Cer, Hex-Cer, and SM but did not capture dihydro species or other lipid classes such as wax esters, cholesterol esters, or phospholipids beyond SM.

#### 2.5.2. Tear Eicosanoid Extraction and Analysis

Tear eicosanoids were extracted from Schirmer strips by UPLC ESI-MS/MS. Strips were placed in tubes containing 4 mL of water and an internal standard (IS) mixture consisting of 10% methanol (400 μL), glacial acetic acid (20 μL), and 20 μL of a deuterated eicosanoid standard mix (1.5 pmol/μL, 30 pmol total; Cayman Chemicals) containing standards such deuterated hydroxyeicosatetraenoic acids (HETEs), prostaglandins, resolvins, leukotrienes, and thromboxanes. Samples and vial rinses (2 mL of 5% methanol) were applied to Strata-X Solid-Phase Extraction (SPE) columns that had been preconditioned with 2 mL of methanol followed by 2 mL of distilled water (dH_2_O). Eicosanoids were eluted with 2 mL of isopropanol, dried under vacuum, and reconstituted in 100 μL of ethanol–distilled water mixture (EtOH:dH_2_O) (50:50) before UPLC ESI-MS/MS analysis.

Eicosanoid separation was performed using a Shimadzu Nexera X2 Liquid Chromatography (LC)-30AD (Shimadzu Corporation, Kyoto, Japan) with a SIL-30AC auto injector and a DGU-20A5R degasser. A reversed-phase LC with an Acentis Express C18 column (150 mm × 2.1 mm, 2.7 µm) was used for eicosanoid separation at 14 min (0.5 mL/min flow rate, 40 °C). This temperature was selected based on method development testing across 35–70 °C, which identified 40 °C as providing an optimal peak shape and chromatographic separation for eicosanoids. Solvent A [acetonitrile:water:formic acid (20:80:0.02, *v*/*v*/*v*)] and Solvent B [acetonitrile:isopropanol:formic acid (20:80:0.02, *v*/*v*/*v*)] were used for this protocol. The column was equilibrated with 100% Solvent A for 5 min, followed by injection of 10 µL of sample. The gradient was as follows: 0–2 min, 100% Solvent A; 2–10.10 min, linear increase to 25% Solvent B at 3 min, 30% Solvent B at 6 min, 55% Solvent B at 6.1 min, 70% Solvent B at 10 min, and 100% Solvent B at 10.10 min; 10.1–13 min, Solvent B maintained at 100%; 13–13.1 min, brought to 100% Solvent A; 13.1–14 min, held constant at 100% Solvent A.

Mass spectrometry detection was performed on an AB Sciex Triple Quad 5500 (SCIEX, Framingham, MA, USA) using MRM negative-ion mode. Q1 and Q3 were set to monitor distinctive precursor and product ion pairs, with fragmentation in Q2 induced by nitrogen collision gas. Instrument parameters were as follows: curtain gas: 20 psi; CAD: medium; ion spray voltage: −4500 V; temperature: 300 °C; gas 1: 40 psi; gas 2: 60 psi. Declustering potential, collision energy, and cell exit potential were varied for each transition. Full MRM transitions and instrument parameters for eicosanoids are provided in the Appendix A.

### 2.6. Plasma Biomarker Analysis

#### 2.6.1. Plasma Sphingolipid Extraction and Analysis

Sphingolipids were extracted and analyzed from 50 μL samples of plasma, following a procedure similar to that described for tear samples, according to published protocols [34].

#### 2.6.2. Plasma Eicosanoid Extraction and Analysis

Eicosanoids were extracted from 50 μL of human plasma and analyzed with liquid chromatography–tandem mass spectrometry (LC-MS/MS) using previously published methods [34,35]. Molecules were separated at a flow rate of 500 μL/min at 50 °C using a 14 min reversed-phase LC method with a Kinetex C18 column (100 × 2.1 mm, 1.7 μm; Phenomenex, Torrance, CA, USA) on a Shimadzu UPLC system (Shimadzu Corporation, Kyoto, Japan). Eluting molecules were analyzed on a hybrid triple quadrupole linear ion trap mass spectrometer (AB 6500 QTRAP; SCIEX, Framingham, MA, USA) using multiple reaction monitoring in negative-ion mode [36,37]. Prostaglandins, thromboxanes, leukotrienes, resolvins, maresins, and protectins were identified based on retention time and *m*/*z* ratio and quantified.

### 2.7. Statistical Analysis

IBM Statistical Package for the Social Sciences (SPSS), Version 30.0.0.0 (Build 172) was used for statistical analyses. Descriptive statistics summarized demographics, comorbidities, medication use, symptom and questionnaire scores, ocular surface and clinical measures, and biomarker data. Sample sizes vary slightly across variables due to occasional missing values. Normality was assessed using the Shapiro–Wilk test. For group comparisons, independent t-tests were applied when normality assumptions were met; otherwise, Mann–Whitney U tests were used. Chi-square tests were used to compare categorical variables such as medication use. To maintain consistency across tables, ocular symptoms and signs are presented as mean (standard deviation [SD]), although non-normally distributed variables were analyzed with Mann–Whitney U tests. To address multiple comparisons, we applied the Benjamini–Hochberg (BH) procedure to control the false discovery rate (FDR) at 10%. When adjusted *p*-values were not available (blanks in the output), the corresponding raw *p*-values were reported, as these are nonsignificant results for which the FDR correction minimally changes the original values.

A forward-selection linear regression model was applied to evaluate whether statistically significant candidate biomarkers (sphingolipids and eicosanoids) predicted GWI case versus control status. For biomarkers measured in both absolute concentration (pmol) and mole percentage, only one representation was retained for regression modeling to avoid redundancy. This led to a panel of 29 unique and statistically significant tear and plasma lipid species. These significant species were entered into exploratory stepwise logistic regression in three batches of 10, 10, and 9 variables. Candidate biomarkers retained from these batches (total of 8) were combined into a single multivariable model to determine which remained significant when tested together. Then, all 8 retained biomarkers were modeled alongside clinical covariates (depression [PHQ-9], PTSD [PCL-M], and meibomian gland parameters) to evaluate their independence from psychiatric and ocular surface measures.

Random forest analysis was performed to identify top predictors for GWI by the rank of variable importance. The discriminative performance of the model for distinguishing GWI cases from controls was evaluated by calculating the area under the receiver operating characteristic (ROC) curve (AUC). All analyses were conducted in R (version 4.5.1) using the randomForestSRC package (version 3.4.2).

## 3. Results

### 3.1. Study Population

A total of 40 Veterans were included, 19 with GWI and 21 Gulf War-era controls. The mean age was similar between groups (GWI: 56.7 ± 5.2 years; controls: 55.7 ± 5.4 years; *p* = 0.57) (Table 1). The groups were also comparable with respect to gender, race, ethnicity, and the prevalence of comorbidities such as diabetes, depression, hypertension, and hyperlipidemia. Before correction for multiple comparisons, GWI cases showed higher rates of PTSD (47% vs. 14%; *p* = 0.02), sleep apnea (68% vs. 24%; *p* = 0.005), and antidepressant use (37% vs. 0%; *p* = 0.002), as well as higher scores for depression (PHQ-9; 12.7 vs. 7.9; *p* = 0.01), fatigue (MFIS; 54.5 vs. 28.4; *p* < 0.001), Widespread Pain Index (9.1 vs. 4.7; *p* = 0.01), and Symptom Severity Scale (8.2 vs. 4.1; *p* < 0.001). After applying the Benjamini–Hochberg procedure to control the false discovery rate (FDR = 10%), only sleep apnea (adjusted *p* = 0.07), antidepressant use (adjusted *p* = 0.04), fatigue (MFIS; adjusted *p* = 0.007), and Symptom Severity Scale (adjusted *p* = 0.007) remained statistically significant.

### 3.2. Ocular Symptoms and Signs in GWI Cases and Controls

Group comparisons revealed differences between GWI and control participants with respect to ocular symptoms (Table 2). Before correction for multiple comparisons, GWI cases had higher Ocular Surface Disease Index (OSDI) scores (40.4 vs. 24.5; *p* = 0.047), Neuropathic Pain Symptom Inventory—Eye (NPSI-Eye) total scores (24.3 vs. 8.9; *p* = 0.02), and Convergence Insufficiency Symptom Survey (CISS) scores (29.3 vs. 11.8; *p* < 0.001) compared with controls. After applying the Benjamini–Hochberg procedure to control the false discovery rate (FDR = 10%), only the CISS remained statistically significant (adjusted *p* = 0.003). In contrast, ocular surface signs were similar across groups.

### 3.3. Tear Biomarkers

Statistical analysis of tear lipid species revealed differences between GWI cases and controls (Table 3). Based on raw *p*-values < 0.05, GWI cases had higher absolute concentrations (pmol) of two ceramides (C14:0, C24:0) and three monohexosyl ceramides (MHCs: C18:0, C22:0, C24:0). When expressed as relative abundances (mol %), GWI cases showed lower levels of several ceramides (dihydroceramide [C16:0 DH], C18:1, C18:0, C20:0, C26:0). In addition to sphingolipids, several eicosanoids were elevated in GWI cases (prostaglandin E2 [PGE2], 15-hydroxyeicosatetraenoic acid [15-HETE], 14(15)-epoxyeicosatrienoic acid [(±)14(15)-EET], 5-oxo-eicosatetraenoic acid [5-OxoETE], 8(9)-epoxyeicosatrienoic acid [(±)8(9)-EET], arachidonic acid [AA]). After correction for multiple comparisons using the Benjamini–Hochberg false discovery rate (FDR = 10%), only (±)14(15)-EET and (±)8(9)-EET remained statistically significant. Complete tear lipid data, including non-significant species, are provided in the Appendix A.

### 3.4. Plasma Biomarkers

Plasma analyses revealed differences in sphingolipids and eicosanoids between GWI cases and controls (Table 4). Based on raw *p*-values < 0.05, absolute concentrations (pmol) of Cer C18:0, Cer C20:0, and dihydrosphingomyelin C16:0 (SM C16:0 DH) were significantly reduced in GWI cases, while SM C20:0 and SM C22:0 were significantly increased. In terms of relative abundance (mol %), GWI cases showed significantly lower levels of MHC C16:0, SM C14:0, SM C16:0, and SM C16:0 DH and higher levels of Cer C22:0, MHC C24:0, SM C20:0, and SM C22:0 compared with controls. In addition to sphingolipids, two plasma eicosanoids were significantly reduced in GWI cases: 11,12-dihydroxyeicosatrienoic acid (11,12-DHET) and 5-oxo-eicosatetraenoic acid (5-OxoETE). After applying the Benjamini–Hochberg procedure to control the false discovery rate (FDR = 10%), the lipid species that remained statistically significant were SM C20:0 (in both absolute concentration and relative abundance; adjusted *p* = 0.09 and 0.002, respectively), MHC C16:0 (adjusted *p* = 0.04), SM C14:0 (adjusted *p* = 0.07), SM C16:0 DH (adjusted *p* = 0.001), and SM C22:0 (adjusted *p* = 0.007). Complete plasma lipid data, including non-significant species, are provided in the Appendix A.

### 3.5. Multivariate Modeling

Multivariate modeling was used to explore which variables best distinguished GWI cases from controls. Out of all tear and plasma sphingolipid and eicosanoid species initially screened, 29 unique species showed significant differences between groups by *t*-test or Mann–Whitney U test analysis using raw *p*-values. These 29 were then entered into exploratory logistic regression in three batches (10, 10, and 9 variables). Across the batches, eight candidate lipid markers were retained: tear eicosanoids 15-HETE and 14,15-EET; tear ceramides C14:0 and C18:0; tear MHC C22:0; plasma ceramide C18:0; and plasma sphingomyelins C16:0 DH and C20:0. When these eight candidates were entered together into a single multivariable model, three lipid species remained: plasma SM C16:0 DH (*p* = 0.071, OR = 168.1, 95% CI 0.65–43,471), plasma SM C20:0 (*p* = 0.052, OR = 0.023, 95% CI 0.001–1.027), and tear eicosanoid 15-HETE (*p* = 0.122, OR ≈ 0.000, 95% CI 0.000–164.2). The reduction from eight to three reflects overlap in lipid pathways, with stepwise regression retaining only the strongest predictors. Finally, when these eight retained species were modeled together with clinical covariates (depression [PHQ-9], PTSD [PCL-M], and meibomian gland parameters), only plasma SM C16:0 DH was consistently retained (*p* = 0.98, OR ≈ 6.9 × 10^237, CI not estimable). Although its effect estimate was unstable, its repeated emergence across both unadjusted and adjusted models highlights its consistency in this pilot study.

A random forest analysis, using both lipid measurements and clinical data, revealed six top predictors in the following order of importance: plasma SM C16:0 DH, plasma SM C20:0, CISS score, Symptom Severity Scale score, MFIS, and plasma SM C 22:0. Notably, both logistic regression and random forest analyses identified plasma SM C16:0 DH and SM C20:0 among the top predictors of GWI status. The random forest model demonstrated strong discriminatory performance between GWI and control participants, with an area under the receiver operating characteristic curve (AUC) of 0.89 (Figure 1). Given the modest sample size, the multivariate and machine learning models should be interpreted as exploratory and hypothesis-generating, and the high AUC likely reflects some degree of overfitting rather than true predictive performance.

## 4. Discussion

In this study, we investigated differences in plasma and tear lipid composition between GWI participants and non-deployed Gulf War-era controls, alongside differences in clinical measures. GWI cases showed a higher prevalence of PTSD and greater use of antidepressant medications. Standardized questionnaires confirmed higher levels of depression (PHQ-9), fatigue (MFIS), and musculoskeletal pain (Widespread Pain Index and Symptom Severity Scale). These findings are consistent with the broader GWI phenotype, in which depression, PTSD, fatigue, and pain are common comorbidities and symptoms thought to reflect underlying neuroimmune dysregulation [2,19]. With respect to ocular outcomes, GWI cases reported significantly greater symptom burden (OSDI, NPSI-E, CISS), despite comparable ocular signs. These results align with prior work demonstrating that Veterans with GWI more frequently report eye symptoms despite similar levels of observable ocular surface signs [4,5].

At the molecular level, lipidomic profiling revealed differences in sphingolipids and eicosanoids in the tears and blood, with 29 unique lipid species differing between GWI cases and controls on univariable testing using raw *p*-values. This broad signal underscores the potential role of lipid pathways in GWI pathophysiology, as these bioactive lipids are central to inflammation and neuronal function [38,39]. After correction for multiple comparisons, a subset of tear eicosanoids ((±)14(15)-EET and (±)8(9)-EET) and plasma sphingolipids (MHC C16:0, SM C14:0, SM C16:0 DH, SM C20:0, and SM C22:0) remained statistically significant. Among these, tear eicosanoid 15-HETE, plasma SM C20:0, and plasma SM C16:0 DH emerged as the most consistent candidate indicators across analytical models, retaining significance even after adjustment for clinical covariates such as depression, PTSD, and meibomian gland parameters—conditions previously associated with eicosanoid and sphingolipid alterations. Random forest analysis identified six top predictors overall, with plasma SM C16:0 DH and SM C20:0 overlapping between logistic regression and random forest models, reinforcing their potential relevance to GWI status.

Although certain ceramide and sphingomyelin differences did not survive multiple comparison correction, the uncorrected patterns provide useful exploratory context when compared with prior studies of lipid dysregulation in GWI and other chronic conditions. Previous metabolic profiling of GWI has highlighted the centrality of lipid dysregulation, including elevated serum SPL such as ceramides and sphingomyelins, as reported in 20 GWI patients vs. 20 non-Veteran healthy controls [14]. In our study, we extend these findings by showing alterations across tears and plasma. On univariable analysis using raw *p*-values, certain tear ceramides were increased (e.g., C14:0, C24:0) while others were reduced, suggesting local remodeling of tear film lipids. In plasma, long-chain ceramides declined (e.g., C18:0, C20:0), whereas very-long-chain sphingomyelins rose (e.g., C22:0). When considered concomitantly, however, plasma SM C16:0 DH was the SPL that remained significantly different between the groups. Interestingly, long- and very-long-chain SPLs have been implicated in other chronic diseases. For example, the ratios of saturated to monounsaturated long-chain ceramide (C18:0/C18:1) and very-long-chain sphingomyelin (C26:0/C26:1) were found to predict prediabetes risk in individuals whose parents had type 2 diabetes (T2D) [40]. Similarly, elevated baseline ceramides (e.g., d18:1–C16:0) were linked to a tenfold higher likelihood of developing Alzheimer’s disease [41]. Although the specific SPL species differ, our findings are consistent with the broader observation that disturbances in sphingomyelin metabolism relate to systemic inflammatory and neurodegenerative changes.

Prior work in GWI has also identified differences in plasma eicosanoid levels. Specifically, decreased levels of plasma prostaglandin F_2_α and leukotriene B_4_ were noted in 37 GWI Veterans vs. 33 healthy controls [42]. In our dataset, exploratory uncorrected analyses suggested potentially higher tear PGE_2_ and lower plasma 11,12-DHET in GWI compared with controls. Although these trends did not persist after correction for multiple comparisons, they remain biologically plausible and align with findings in prior studies. PGE2 is a well-established mediator of inflammation [43,44], and prior work in dry eye disease (DED) has suggested that elevated levels of PGE2 may help sustain ocular surface inflammation and nociceptor excitation [45]. Thus, even as preliminary observations, elevated tear PGE2 in GWI may offer one plausible explanation for the mismatch between heightened ocular symptoms in cases despite comparable ocular signs between cases and controls.

The cytochrome P450 (CYP450)–soluble epoxide hydrolase (sEH) pathway that generates and metabolizes epoxyeicosatrienoic acids (EETs) is another critical regulator of inflammation and vascular tone [46]. EETs, derived from arachidonic acid via CYP450 epoxygenases, possess vasodilatory and anti-inflammatory properties and are rapidly hydrolyzed by sEH into less active dihydroxyeicosatrienoic acids (DHETs) [46]. The exploratory trend toward reduced plasma 11,12-DHET in GWI is consistent with potential alterations in this pathway, although this finding did not survive multiple comparison correction. After correction for multiple comparisons (FDR = 10%), two tear eicosanoids, (±)14(15)-EET and (±)8(9)-EET, remained significantly elevated in GWI. These findings show dysregulation of the CYP–EET–sEH axis in GWI, suggesting that elevated tear EETs may represent a localized compensatory response to inflammation. These findings support lipidomics as a promising avenue for biomarker discovery in GWI.

Our study must be considered in light of its limitations. First, the pilot cohort was modest in size, which limited the statistical power and resulted in wide confidence intervals in our regression model. Therefore, all multivariate and machine learning results should be interpreted as exploratory rather than predictive. The random forest AUC of 0.89 likely reflects overfitting inherent to a limited dataset, and these findings should be viewed as hypothesis-generating until replicated in larger cohorts. Second, the cross-sectional nature of our design prevents conclusions about whether lipid variations are causal drivers of GWI pathophysiology, compensatory changes, or consequences of illness chronicity. Accordingly, the associations observed in this study should be interpreted as non-causal and exploratory. Third, some variables had missing data, and thus analyses were conducted with available cases, which reduced the effective sample size for certain comparisons and may have lowered statistical power. Fourth, our cases were Veterans deployed to the Gulf who met criteria for Gulf War Illness, while our controls were non-deployed Gulf War-era Veterans without GWI. Thus, differences between groups may reflect not only GWI status but also background differences associated with deployment. Finally, the GWI group had a higher burden of comorbidities such as depression, as well as greater antidepressant use, which are known to influence lipid biology and may introduce additional confounding [47,48]. These effects have also been described in psychiatric conditions such as post-traumatic stress disorder, which was more prevalent in the GWI group [49,50]. As an additional sensitivity analysis, we repeated univariable lipid species comparisons after excluding individuals taking antidepressants or statins. The overall patterns were similar, but no lipid species remained significant after correction for multiple comparisons, which is expected given the reduced sample size. These considerations highlight the need for larger and more closely matched cohorts in future studies. Despite these limitations, this study provides evidence that lipidomics can identify compartment-specific changes in GWI, including tear PGE2, plasma 11,12-DHET, and plasma SM C16:0 DH as candidate biomarkers. These results support the importance of lipid pathways contributing to the neuroimmune disturbances of GWI and highlight the potential value of tear sampling for biomarker discovery.

The logical next step of this research is to further investigate these preliminary lipid signals in larger and better-matched cohorts (including non-deployed individuals with GWI symptoms and deployed individuals without GWI symptoms) to determine their reproducibility, stability, and clinical relevance. Independent validation will be essential before any lipid species can be considered for diagnostic or therapeutic applications. Nonetheless, refining lipid signatures may eventually help advance objective disease identification but also provide mechanistic insight into potentially targetable biological pathways. This is particularly significant given that several modulators of sphingolipid biosynthesis and signaling are already available, and one such agent—fingolimod—has been shown to prevent memory loss in a toxin-induced animal model of GWI [51]. Together, these findings underscore how lipid biomarker discovery can bridge basic and translational research, accelerating the path toward precision diagnostics and mechanism-based interventions that improve care and outcomes for Veterans with GWI. Furthermore, our findings highlight the potential of tear fluid as a minimally invasive and practical source for biomarker discovery.

## 5. Conclusions

In this pilot study, we demonstrated increased ocular symptom burden despite similar ocular signs in GWI cases compared to non-deployed GW-era controls. We identified differences in tear and plasma lipid profiles among the two groups, particularly alterations in eicosanoids and sphingolipids. While the modest sample limits generalizability, our findings highlight the potential of lipidomics for biomarker development in GWI, with tear fluid emerging as a minimally invasive sample for biomarker exploration. Larger cohorts are necessary in future studies to validate our findings and further explore disease pathogenesis, with the ultimate goal of improving the diagnosis and management of GWI.

## Figures and Tables

**Figure 1 biomolecules-15-01716-f001:**
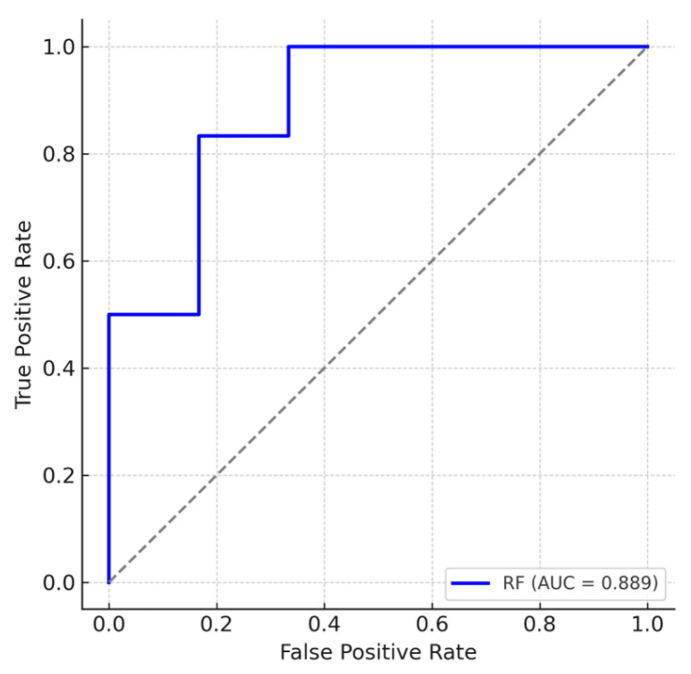
Receiver operating characteristic (ROC) curve for the random forest model classifying Gulf War Illness (GWI) cases (n = 19) versus controls (n = 21) using biomarker and clinical data. The true positive rate (sensitivity) is plotted against the false positive rate (1–specificity). The blue line represents the random forest classifier, and the gray dashed line represents random chance (AUC = 0.5). The model achieved an area under the curve (AUC) of 0.89, indicative of a strong discriminatory performance.

**Table 1 biomolecules-15-01716-t001:** Demographics, comorbidities, medication use, and systemic symptoms in GWI cases and controls.

Characteristic	GWI (n = 19)	Controls (n = 21)	Raw *p*-Value	Adjusted *p*-Value (FDR)
**Demographics**				
Age, mean ± SD (years)	56.7 ± 5.2	55.7 ± 5.4	0.57	0.72
Sex, male % (n)	89.5% (17)	85.7% (18)	1.00	1.00
Race, White % (n)	63.2% (12)	42.9% (9)	0.15	0.28
Race, Black % (n)	31.6% (6)	57.1% (12)		
Race, Asian % (n)	5.3% (1)	0% (0)		
Ethnicity, Hispanic % (n)	36.8% (7)	38.1% (8)	0.94	0.96
**Comorbidities % (n)**				
Diabetes mellitus	21.1% (4)	19.0% (4)	1.00	0.93
PTSD	47.4% (9)	14.3% (3)	0.02	0.19
Depression	36.8% (7)	23.8% (5)	0.37	0.57
Sleep apnea	68.4% (13)	23.8% (5)	0.005	0.07
Hypertension	31.6% (6)	61.9% (13)	0.055	0.26
Hyperlipidemia	42.1% (8)	52.4% (11)	0.52	0.70
**Medications % (n)**				
Ocular medications	47.4% (9)	33.3% (7)	0.37	0.57
NSAIDs	63.2% (12)	33.3% (7)	0.06	0.26
ASA	21.1% (4)	14.3% (3)	0.57	0.74
Fish oil	36.8% (7)	19.0% (4)	0.21	0.44
Statins	36.8% (7)	42.9% (9)	0.70	0.83
Antidepressants	36.8% (7)	0% (0)	0.002	0.04
Antianxiety	26.3% (5)	14.3% (3)	0.34	0.56
**Systemic Symptoms, mean (SD)**				
Depression (PHQ-9)	12.7 (5.1)	7.9 (7.4)	0.01	0.25
PTSD (PCL-M)	47.9 (19.1)	37.7 (16.5)	0.13	0.36
Insomnia (PSQI)	13.2 (4.5)	10.6 (5.5)	0.11	0.32
Fatigue (MFIS)	54.5 (14.5)	28.4 (22.7)	<0.001	0.007
Widespread Pain Index	9.1 (5.2)	4.7 (4.0)	0.01	0.17
Symptom Severity Scale	8.2 (2.0)	4.1 (2.8)	<0.001	0.007
SF-12 physical score	32.5 (9.7)	40.2 (13.1)	0.12	0.35
SF-12 mental score	39.4 (16.3)	47.3 (11.1)	0.17	0.39

GWI = Gulf War Illness; SD = standard deviation; PTSD = post-traumatic stress disorder; NSAID = nonsteroidal anti-inflammatory drug; ASA = aspirin; PHQ-9 = Patient Health Questionnaire-9; PTSD (PCL-M) = Post-Traumatic Stress Disorder Checklist—Military version; PSQI = Pittsburgh Sleep Quality Index; MFIS = Modified Fatigue Impact Scale; SF-12 = 12-Item Short-Form Survey for quality of life. Raw *p* < 0.05 was considered statistically significant. Adjusted *p*-values were calculated using the Benjamini–Hochberg procedure to control the false discovery rate (FDR = 10%). Variables with adjusted *p* ≤ 0.10 were considered significant.

**Table 2 biomolecules-15-01716-t002:** Comparison of ocular symptoms and signs between GWI cases and controls.

Measure	GWI Mean (SD) (n = 19)	Control Mean (SD) (n = 21)	Raw *p*-Value	Adjusted *p*-Value (FDR)
**Ocular Symptoms (Questionnaires)**				
OSDI	40.4 (25.0)	24.5 (22.5)	0.047	0.23
DEQ-5	9.6 (4.1)	7.4 (5.1)	0.14	0.36
NPSI-Eye total score	24.3 (21.3)	8.9 (13.3)	0.02	0.13
CISS	29.3 (11.5)	11.8 (11.7)	<0.001	0.003
**Ocular Pain Ratings**				
Right eye (OD), now	2.1 (2.7)	0.6 (0.9)	0.19	0.19
Left eye (OS), now	2.5 (3.3)	1.0 (1.4)	0.28	0.26
Right eye (OD), average 1 week	2.1 (2.3)	1.0 (1.3)	0.15	0.27
Left eye (OS), average 1 week	2.6 (2.8)	1.1 (1.3)	0.12	0.23
Right eye (OD), worst 1 week	2.9 (3.1)	1.1 (1.5)	0.10	0.19
Left eye (OS), worst 1 week	3.2 (3.3)	1.2 (1.5)	0.09	0.19
Pre-anesthetic OD	2.2 (2.5)	1.1 (2.4)	0.11	0.36
Pre-anesthetic OS	2.0 (2.5)	1.0 (2.2)	0.13	0.38
Post-anesthetic OD	1.0 (2.3)	0.3 (0.9)	0.67	0.42
Post-anesthetic OS	1.1 (2.2)	0.3 (0.9)	0.50	0.39
**Clinical Signs**				
InflammaDry (MMP) OD	1.4 (1.0)	0.8 (0.9)	0.07	0.27
InflammaDry (MMP) OS	0.8 (0.8)	0.8 (0.8)	0.90	0.92
TBUT OD (seconds)	9.3 (3.9)	9.6 (5.1)	0.86	0.92
TBUT OS (seconds)	9.4 (4.7)	9.5 (5.0)	0.95	0.97
Corneal staining OD	1.2 (2.0)	1.8 (2.6)	0.47	0.61
Corneal staining OS	1.5 (2.1)	1.8 (3.3.)	0.69	0.85
Schirmer OD (mm/5 min)	16.1 (8.3)	17.0 (10.9)	0.78	0.78
Schirmer OS (mm/5 min)	17.8 (7.2)	20.1 (12.1)	0.73	0.73
Eyelid vascularity OD	0.5 (0.7)	0.2 (0.4)	0.41	0.42
Eyelid vascularity OS	0.5 (0.7)	0.2 (0.4)	0.41	0.42
Meibum quality OD	1.1 (1.0)	0.8 (0.4)	0.34	0.36
Meibum quality OS	1.2 (1.0)	1.0 (0.9)	0.46	0.68
MG gland dropout OD	1.9 (1.1)	1.3 (0.9)	0.08	0.28
MG gland dropout OS	2.2 (1.3)	1.5 (0.8)	0.13	0.27

GWI = Gulf War Illness; SD = standard deviation; OD = right eye; OS = left eye; OSDI = Ocular Surface Disease Index; DEQ-5 = Dry Eye Questionnaire-5; NPSI = Neuropathic Pain Symptom Inventory; CISS = Convergence Insufficiency Symptoms Survey; MMP = Matrix Metalloproteinase; TBUT = tear breakup time. MG = meibomian gland. Raw *p* < 0.05 was considered statistically significant. Adjusted *p*-values were calculated using the Benjamini–Hochberg procedure to control the false discovery rate (FDR = 10%). Variables with adjusted *p* ≤ 0.10 were considered significant.

**Table 3 biomolecules-15-01716-t003:** Tear sphingolipids and eicosanoids in GWI cases and controls.

Biomarkers	GWI Mean (SD) (n = 19)	Control Mean (SD) (n = 21)	*p*-Value	Adjusted *p*-Value (FDR)
**Tear Sphingolipids (pmol)**				
Cer C14:0	0.33 (0.30)	0.16 (0.10)	0.046	0.23
Cer C24:0	3.70 (3.75)	1.75 (1.49)	0.04	0.26
MHC C18:0	0.10 (0.08)	0.05 (0.03)	0.03	0.19
MHC C22:0	0.24 (0.20)	0.12 (0.07)	0.02	0.17
MHC C24:0	0.42 (0.39)	0.20 (0.11)	0.02	0.55
**Tear Sphingolipids (mol %)**				
Cer C16:0 DH	1.31 (0.52)	1.72 (0.56)	0.01	0.23
Cer C18:1	2.05 (1.03)	2.89 (1.38)	0.04	0.25
Cer C18:0	4.42 (1.14)	5.45 (1.15)	0.01	0.17
Cer C20:0	4.02 (1.14)	4.88 (0.82)	0.01	0.17
Cer C26:0	5.20 (3.33)	7.72 (4.02)	0.03	0.22
**Tear Eicosanoids (pmol/mL)**				
PGE2	0.01 (0.01)	0.004 (0.003)	0.04	0.19
15-HETE	0.12 (0.15)	0.04 (0.04)	0.02	0.19
(±)14(15)-EET	0.04 (0.03)	0.01 (0.01)	0.002	0.04
5-OxoETE	0.38 (0.35)	0.20 (0.21)	0.02	0.25
(±)8(9)-EET	0.03 (0.03)	0.01 (0.01)	0.008	0.09
AA	29.83 (22.64)	15.70 (15.93)	0.02	0.29

GWI = Gulf War Illness; SD = standard deviation; Cer = Ceramide (chain length indicated by Cxx and saturation status by “:x”, where 0 = saturated and 1 = monoun-saturated); MHC = Monohexosylceramide (chain length indicated by Cxx and saturation status by “:x”, where 0 = saturated and 1 = monoun-saturated); DH = dihydro form of the lipid species (saturated sphingoid base); PGE2 = Prostaglandin E2; 15-HETE = 15-Hydroxyeicosatetraenoic acid; (±)14(15)-EET = 14(15)-Epoxyeicosatrienoic acid (racemic mixture); 5-OxoETE = 5-Oxo-Eicosatetraenoic acid; (±)8(9)-EET = 8(9)-Epoxyeicosatrienoic acid; AA = Arachidonic Acid; pmol = picomoles; mol % = mole percent; pmol/mL = picomoles per milliliter. Raw *p* < 0.05 was considered statistically significant. Adjusted *p*-values were calculated using the Benjamini–Hochberg procedure to control the false discovery rate (FDR = 10%). Variables with adjusted *p* ≤ 0.10 were considered significant.

**Table 4 biomolecules-15-01716-t004:** Plasma sphingolipids and eicosanoids in GWI cases and controls.

Biomarkers	GWI Mean (SD) (n = 19)	Control Mean (SD) (n = 21)	*p*-Value	Adjusted *p*-Value (FDR)
**Plasma Sphingolipids (pmol)**				
Cer C18:0	8.82 (3.12)	11.84 (4.39)	0.02	0.18
Cer C20:0	20.83 (6.58)	26.16 (9.49)	0.048	0.25
SM C16:0 DH	537.32 (68.77)	622.58 (127.11)	0.01	0.17
SM C20:0	1334.88 (229.70)	1152.93 (157.26)	0.005	0.09
SM C22:0	2175.69 (452.10)	1911.47 (262.31)	0.03	0.19
**Plasma Sphingolipids (mol %)**				
Cer C22:0	18.47 (2.33)	16.78 (2.30)	0.01	0.19
MHC C16:0	15.14 (2.28)	17.89 (2.78)	0.002	0.04
MHC C24:0	39.72 (5.50)	36.28 (3.56)	0.02	0.36
SM C14:0	8.09 (0.75)	10.10 (2.76)	0.005	0.07
SM C16:0	4.24 (0.70)	4.67 (0.61)	0.048	0.25
SM C16:0 DH	4.75 (0.31)	5.44 (0.47)	<0.001	0.001
SM C20:0	11.74 (1.00)	10.19 (0.95)	<0.001	0.002
SM C22:0	19.05 (1.98)	16.87 (1.26)	<0.001	0.007
**Plasma Eicosanoids (pmol/mL)**				
(±)11,12-DHET	1.18 (0.30)	1.38 (0.30)	0.04	0.23
5-OxoETE	31.28 (13.30)	46.38 (29.92)	0.05	0.25

GWI = Gulf War Illness; SD = standard deviation; Cer = ceramide (chain length indicated by Cxx and saturation status by “:x”, where 0 = saturated and 1 = monounsaturated); SM = sphingomyelin (chain length indicated by Cxx and saturation status by “:x”, where 0 = saturated and 1 = monounsaturated); DH = dihydro form of the lipid species (saturated sphingoid base); MHC = monohexosylceramide (chain length indicated by Cxx and saturation status by “:x”, where 0 = saturated and 1 = monounsaturated); (±)11,12-DHET = 11,12-dihydroxyeicosatrienoic acid (racemic mixture); 5-OxoETE = 5-Oxo-eicosatetraenoic acid; pmol = picomoles; mol % = mole percent; pmol/mL = picomoles per milliliter. Raw *p* < 0.05 was considered statistically significant. Adjusted *p*-values were calculated using the Benjamini–Hochberg procedure to control the false discovery rate (FDR = 10%). Variables with adjusted *p* ≤ 0.10 were considered significant.

## Data Availability

The data presented in this study are available in the article and Appendix A.

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
