# Peer review of "Investigating the Eye as a Biomarker of Gulf War Illness: Sphingolipid and Eicosanoid Composition in Tears and Plasma"

_biomolecules, 2025, doi:10.3390/biom15121716_

Round 1

Reviewer 1 Report

Comments and Suggestions for Authors

Dear authors,

Thank you for your submission, and please accept my apologies for the late reply.

I have several comments regarding the analytical path.

  1. Tear samples were collected using the Schirmer strip. Did you also analyze blank Schirmer strips to obtain the eventual background?
  2. The quantitation of lipids is described, and results are presented, but I could not identify the calibration method, the calibration curve, or any calibration data. Was the calibration curve obtained?
  3. The MRM method used was described in previous publications, and you have cited the prior work; however, it would be of interest to readers and peers alike to have an overview of the MRM method, e.g., as a tabular listing of precursors and fragments.
  4. Could you explain why the LC methods for eiconosaids and the sphingolipids were performed at different column temperatures, i.e., 40°C and 60°C?
  5. The study limitations were mentioned; however, could you comment on why there were no veterans included as a control group who were not deployed, did not suffer from GWI, but had the same ophthalmological symptoms? Would that not have been a better control group?

Kind regards.

Reviewer 2 Report

Comments and Suggestions for Authors

The manuscript “Investigating the Eye as a Biomarker of Gulf War Illness: Sphingolipid and Eicosanoid Composition in Tears and Plasma” is a well-conceived and technically sophisticated pilot study exploring lipidomic alterations in Veterans affected by Gulf War Illness (GWI). The integration of systemic and ocular lipid profiling represents an innovative approach and fits well within the biomarker discovery aims of Biomolecules. The topic is clinically relevant, as ocular symptoms are increasingly recognized as part of the GWI phenotype, and the use of tear fluid as a minimally invasive biological sample is both practical and original. However, the strength of the conclusions is limited by several methodological and interpretive issues that should be addressed before publication.

The main limitation concerns the small sample size (n=19 GWI, n=21 controls), which makes all multivariate and machine-learning analyses exploratory rather than confirmatory. The reported AUC of 0.89 is impressive but likely reflects overfitting in such a restricted dataset. This limitation should be clearly stated, and the authors should describe the model as hypothesis-generating rather than predictive. Similarly, the cross-sectional design does not allow any inference on causality. It remains unclear whether the lipid alterations observed are causal, compensatory, or secondary to chronic disease and comorbidities. This point should be emphasized more explicitly in the Discussion.

The composition of the control group also deserves clarification. Controls were Gulf War–era Veterans who were not deployed, rather than deployed asymptomatic Veterans or civilian controls. This choice introduces potential selection bias, as the groups may differ in terms of exposures, stress levels, and medications. Moreover, the GWI group showed higher prevalence of PTSD, depression, and antidepressant use, all of which are known to affect lipid metabolism. It would be useful to clarify whether these factors were adjusted for in the statistical models and to discuss their possible influence on sphingolipid and eicosanoid pathways. If available, an analysis excluding individuals on antidepressants or statins would strengthen the argument that the observed differences are related to GWI itself.

From a statistical perspective, only a few lipid species remained significant after false discovery rate correction, whereas the Discussion interprets several others as meaningful. The narrative should focus on those molecules that survived multiple comparison correction and refer to the rest as preliminary trends. The authors might also consider including a correlation analysis between lipid levels and clinical scores (OSDI, MFIS, PHQ-9, etc.) to enhance the biological plausibility of the associations.

Some technical details are missing regarding the analytical performance of the LC–MS/MS platform. It would improve transparency to include intra- and inter-assay coefficients of variation, limits of detection, and quality control procedures in the Supplementary Materials. Additionally, the figures should clearly indicate sample sizes and maintain high resolution suitable for publication.

In terms of language and tone, the manuscript is generally clear and well written, but certain expressions are too strong for a pilot study. Terms such as “biomarker” or “diagnostic indicator” should be replaced by “potential indicator” or “candidate molecule,” and conclusions should be phrased cautiously to reflect the exploratory nature of the work. The final paragraph of the Discussion could be softened to stress that these results support further investigation rather than immediate clinical translation.

Round 2

Reviewer 2 Report

Comments and Suggestions for Authors

The authors have addressed all the concerns raised in the previous review round with clarity and precision. The methodological clarifications, the strengthened discussion of limitations, and the more cautious wording are appropriate for a pilot study and substantially improve the manuscript. I have no further comments. The work is now suitable for publication.

Author Response

Comment 1: The authors have addressed all the concerns raised in the previous review round with clarity and precision. The methodological clarifications, the strengthened discussion of limitations, and the more cautious wording are appropriate for a pilot study and substantially improve the manuscript. I have no further comments. The work is now suitable for publication.

Response 1: We thank the reviewer for their re-evaluation of the manuscript and for the positive feedback. We appreciate the reviewer’s time and are pleased that the revisions have satisfactorily addressed all concerns.